# Mechanisms of the Antitumor Activity of Low Molecular Weight Heparins in Pancreatic Adenocarcinomas

**DOI:** 10.3390/cancers12020432

**Published:** 2020-02-13

**Authors:** Alexandros Bokas, Pavlos Papakotoulas, Panagiotis Sarantis, Adriana Papadimitropoulou, Athanasios G Papavassiliou, Michalis V Karamouzis

**Affiliations:** 11st Department of Medical Oncology, Theagenion Hospital, 54007 Thessaloniki, Greece; alexanderbokas@outlook.com (A.B.); papakotoulas@gmail.com (P.P.); 2Molecular Oncology Unit, Department of Biological Chemistry, Medical School, National and Kapodistrian University of Athens, 11527 Athens, Greece; panayotissarantis@gmail.com (P.S.); papavas@med.uoa.gr (A.G.P.); 3Center of Basic Research, Biomedical Research Foundation of the Academy of Athens, 11527 Athens, Greece; adapapadim@gmail.com; 4First Department of Internal Medicine, ‘Laiko’ General Hospital, Medical School, National and Kapodistrian University of Athens, 11527 Athens, Greece

**Keywords:** LMWH, immunotherapy, PDAC, tumor microenvironment

## Abstract

Immune checkpoint inhibitors have revolutionized cancer treatment in the last decade. Despite the progress in immunotherapy, most pancreatic cancer patients still do not derive benefit when receiving immune-based therapies. Recently, resistance mechanisms to immune therapies have been mainly focused on tumor microenvironment properties. Pancreatic cancer is considered one of the most lethal and difficult to treat tumors due to its highly immunosuppressive and desmoplastic microenvironment. Low molecular weight heparins (LMWHs) have been used for the treatment and prevention of thromboembolic disease in these patients. However, many nonanticoagulant properties attributed to LMWHs have been described. Exploiting LMWH properties in a combined treatment modality with immune checkpoint inhibition and chemotherapy could provide a new approach in the management of pancreatic adenocarcinoma patients. The ability of LMWH to interfere with various aspects of the tumor microenvironment could result in both the alleviation of immunosuppression and improvement in drug delivery within the tumor, leading to higher cancer cell destruction rates and more potent immune system activity that would, ultimately, lead to better patient outcomes.

## 1. Introduction

Cancer therapy has traditionally been based on targeting tumor cells with cytotoxic chemotherapy or radiotherapy, while surgery is the prevailing modality for long-term disease control and cure [1]. Recently, there has been an increase in the survival rates and long-term benefits due to immunotherapy-based treatments with the use of checkpoint inhibitors [2]. 

However, despite its proven efficacy in the treatment of many solid and hematological cancers, a significant proportion of patients do not derive benefit from the administration of immunotherapy [3].

The exact mechanism of resistance to immunotherapy is still under investigation, but recent data indicate that the tumor microenvironment plays a major role in establishing immune-suppressive phenotypes, rendering cancers refractory to treatment. One such case is pancreatic adenocarcinoma (PDAC), where immunotherapy has shown only minimal benefit in a very small subgroup of patients bearing the microsatellite instability (MSI) high phenotype [4]. Current research on PDAC focuses on the combination of immunotherapy with other therapeutic modalities to reverse immunosuppression and accomplish efficient tumor inhibition [5].

PDAC has been recognized as a highly thrombogenic tumor, with over 20% of patients suffering from venous thromboembolism (VTE) during the course of their disease; VTE is considered to be the second leading cause of death in these patients [6]. Low molecular weight heparins (LMWHs) are recommended as a first-line option for the treatment and primary prophylaxis of VTE in pancreatic cancer patients [7,8,9]. LMWHs, such as dalteparin, tinzaparin, and enoxaparin, are anticoagulants that inhibit the final step in the common pathway of the coagulation cascade (they inhibit factor Xa by activating antithrombin III) [10]. They constitute the first-line treatment for VTE patients, as they are more beneficial compared to unfractionated heparin, due to the reduced administration frequency that comes as a result of their extended half-lives. Additionally, they are related to the reduced incidence of heparin-induced thrombocytopenia (HIT) compared to unfractionated heparin UFH, and they can be administered through subcutaneous injection, which can be easily performed by the patients themselves [11]. 

Moreover, there have been numerous studies with experimental models such as cell lines, tissues, and mice in various types of cancers that have demonstrated the antitumor and antimetastatic effect LMWHs possess [12,13,14]. Furthermore, there have been clinical trials for small cell lung cancer (SCLC) and breast cancer where the overall survival of patients has been prolonged after the use of LMWHs [15,16,17]. These findings need to be further studied to determine whether LMWHs retain antitumor activity in addition to their well-established role as anticoagulants. Based on the combinatorial therapy approach to treat a highly malignant and refractory cancer such as PDAC, we propose that LMWHs could augment the antitumor effectiveness of immune checkpoint inhibitors and chemotherapy, and we outline the potential mechanisms by which this could be achieved.

## 2. The Pancreatic Ductal Adenocarcinoma Microenvironment

Current knowledge supports the notion that PDAC consists of a minority of neoplastic epithelial cells surrounded by a high-density desmoplastic stroma that accounts for 60–90% of the total tumor mass [18]. The presence of desmoplasia is a hallmark in PDAC pathogenesis and progression and can be recognized mainly at the primary tumor site [19,20]. The term tumor microenvironment (TME) is used to describe an assembly of elements that form pancreatic neoplasia and consists of tumor cells, various nonmalignant stromal cells, and active factors as shown in Figure 1 [21]. The tumor microenvironment is a dynamic entity, and this may account for the diversity observed in different tissues; however, main features such as desmoplasia, abnormal vascularity, and fibroblast activation are almost always present [22,23,24]. The most prominent nontumor components include mesenchymal cells, such as activated pancreatic stellate cells and fibroblasts, inflammatory cells, immune cells, blood vessels, and extracellular matrix (ECM) rich in proteins and growth factors [25]. The TME exerts a multitude of actions ranging from tumor formation, cancer dissemination, and resistance to therapy.

Fibroblasts are the predominant cell type within the TME where they form a heterogeneous group of cells that contribute to the formation of ECM and support tumor growth [26]. Pancreatic stellate cells represent a subset of cancer-associated fibroblasts that exist in a quiescent fat-storing state in healthy pancreatic tissue. Once activated, i.e., as the result of inflammation, injury, or tumor formation, they adopt a myofibroblast-like phenotype with high proliferating capacity and increased secretion of extracellular matrix proteins [27]. 

Immune cells resident in the TME are mostly myeloid-derived suppressor cells (MDSCs), tumor-associated macrophages (TAMs), and tumor-infiltrating lymphocytes. Regulatory T cells (Tregs) have the capacity to highly infiltrate various tumor types such as pancreatic cancer, and advanced cancers often display higher levels of Treg accumulation. Treg elevation inside the tumors is performed via transforming growth factor beta TGFb, which induces the differentiation of naïve CD4+ T cells into Tregs. In addition, chemokines produced by the tumor play an important role; one example is CCL22, which is able to stimulate CCR4. Tregs suppress the T cell immune responses of antigen-presenting cells, including dendritic cells (DCs) and macrophages. Tregs can kill effector T cells within tumors through the FasL–Fas signaling pathway as well as via granzyme B- and perforin-mediated cytotoxicity, through the reduction of interleukin (IL)-2 and the differentiation/maturation of DCs [28].

The TME is further characterized by intense hypoxia, and when this is combined with vascular compression caused by desmoplasia, it triggers the process of angiogenesis in order to support the continuous nutrient requirement of the tumor [29]. The essential elements of ECM are collagen, fibronectin, proteoglycans, hyaluronic acid, active enzymes, and proteinases that accumulate and transform normal tissue into desmoplastic neoplasia with vascular and lymphatic abnormalities [25]. The TME represents a dynamic entity that can modify its composition, particularly in the earlier stages of carcinogenesis. Tumor progression to advanced stages and the increase of the ability to metastasize coincides with a process called epithelial-to-mesenchymal transition (EMT). EMT is a process where epithelial cells gradually convert to mesenchymal cells with motile features that enable them to migrate to distant sites. Once relocated, EMT is reversed and tumor cells grow and set up a microenvironment similar to the primary site [29]. Normal pancreatic tissue has the potential to regenerate after injury in a process involving the activation of stellate cells and fibrosis that isolates the affected area, protects healthy tissue, and resolves after tissue repair. Cancer cells disrupt this regenerative procedure by causing the continuous activation of stellate cells with subsequent excessive stroma production, contributing to tumor growth and disease progression [30]. 

Invasive pancreatic cancer evolves in a step-wise pattern from precursor lesions, most commonly initiated by pancreatic intraepithelial neoplasia. Early genetic events include the mutational activation of oncogenes, with *K-ras* being the most frequent, and the inactivation of tumor suppressor genes, such as p16, p53, and Smad4 [31,32]. Neoplastic and stromal cells coexist and cooperate in tumor growth and metastasis, exploiting a variety of cell-to-cell and cell-to-stroma interactions mediated by a number of cytokines and growth factors. This interplay constitutes a major contributor to the resistance to therapy frequently observed in pancreatic cancer [33].

Pancreatic cancer is considered an incurable malignancy with currently approved chemotherapeutics offering only minimal survival benefit; and immune-based approaches failing to produce satisfactory responses [34]. The high refractory nature of PDAC is attributed mainly to the TME; therefore, targeting the microenvironment has become a front-line research focus. However, most of the efforts performed had little effect or even worsened the outcomes on preclinical models. Current research focuses on combination therapy targeting both tumor and stromal cells [35]. Our hypothesis is based on the efficiency of standard chemotherapy with immune checkpoint inhibitors, while adding LMWH as an adjunct in order to achieve the normalization of the microenvironment. Τhis should ultimately lead to the destruction of cancer cells, a subsequent release of tumor-associated antigens, and an increase of the TME’s infiltration by effective immune cells.

## 3. The Role of LMWHs in Pancreatic Cancer

Beyond their role in decreasing VTE occurrence in pancreatic cancer patients, LMWHs may help to increase survival by affecting tumor progression, metastasis formation, and angiogenesis [26,36,37]. Our hypothesis is based on the specific actions of LMWHs that can affect circulating tumor cells and the TME in order to achieve a better response to chemotherapeutics and checkpoint inhibitors.

### 3.1. Effect on Heparan Sulfate Proteoglycans/Heparanase

Heparan sulfate proteoglycans (HSPGs) are conjugates of heparan sulfate and amino acids with a critical role in extracellular matrix protein synthesis and integrity through various interactions with other elements of ECM and plasma membranes. The heparan sulfate chains, due to their broad structural diversity, are able to bind and interact with a wide variety of proteins, such as growth factors, chemokines, and enzymes in the TME, regulating the availability and action of these molecules [38,39]. Heparanase is an endoglycosidase rarely expressed in normal tissue, while it is overexpressed in pancreatic tumors [40]. It acts by cleaving heparan sulfate side chains from HSPGs, which results in both the dismantlement of ECM and the release of angiogenesis promoting growth factors, such as vascular endothelial growth factor A (VEGF-A) and fibroblast growth factor 2 (FGF-2). Thus, heparanase activity is associated with tumor invasion through the degradation of HSPGs and ECM and with angiogenesis due to growth factor release [41]. Heparanase has also been found to have a nonenzymatic procoagulant activity that is mediated by inducing tissue factor (TF) expression and dissociating tissue factor pathway inhibitor (TFPI) from the cell surface [42].

Although LMWHs have never been evaluated in regards to their direct effects on heparanase, they might exhibit a suppressing activity, taking into account that heparin is known to inhibit heparanase [43] and that LMWHs are potent inhibitors of aggreganases 1 and 2, which are also involved in extracellular matrix degradation [13]. The activity of LMWHs is expressed through the inhibition of VEGF-A and FGF-2 as well as the increased release of TFPI (both are presented in Figure 2), suggesting an antagonistic role of LMWHs in both the angiogenetic and the procoagulant activity of heparanase [12]. 

### 3.2. Effect on Metastasis Formation

LMWHs exert a multitude of actions that can compromise the metastatic potential of pancreatic cancer and influence responsiveness to therapy. A vital component of the metastatic potential is the cancer cells detaching from the tumor and entering the circulation. There, they attract platelets and form a tumor cell–platelet clot that helps malignant cells evade immune recognition and attack by immune cells [36]. This tumor cell–platelet aggregate formation is mediated by P-selectin, followed by L-selectin, which drives the recruitment of leucocytes, further resulting in shaping tumor cells containing thrombi that eventually reach a distant site of vascular arrest; tumor cells then activate endothelial cells, which in turn stimulate the expression of E-selectin that also mediates metastasis formation.

An alternative mediator of tumor cell–platelet interplay, although not prevailing over P-selectin, is through platelet integrin aΙΙb3, which interacts with integrin ανβ3 on tumor cells, thus contributing to tumor thrombi vascular arrest [44]. The formation of metastatic foci requires the homing of bone marrow progenitor cells that is dependent on the presence of α4β1 (VLA-4)(very late antigen-4) integrins and the subsequent binding of vascular cell adhesion molecule 1 (VCAM-1) and fibronectin [45,46].

Another pathway that contributes to metastasis formation in pancreatic cancer is the C-X-C motif chemokine ligand 12 (CXCL12)/C-X-C chemokine receptor type 4 (CXCR4) axis. PDAC cells overexpress CXCR4, and CXCL12 is expressed in target organs attracting malignant cells to promote metastasis formation [47]. Forming a paracrine loop of the VEGF-mediated expression of CXCR4 by endothelial cells and the CXCL12-induced expression of VEGF by endothelial cells [47], this pathway is believed to assist in pancreatic cancer progression by inducing angiogenesis and lymphangiogenesis [48].

LMWHs inhibit P- and L-selectin and VLA-4/VCAM-1-mediated cellular interactions, as shown in Figure 3, thereby decreasing metastatic formation [13]. Selectins are vascular cell adhesion molecules that recognize specific sialyl Lewis^x/a^ carrying glycoproteins that exist in normal leukocytes and endothelium. Tumor cell surface molecules carrying sialyl Lewis^x/a^ can also be distinguished from selectins, mediating tumor cell interactions with platelets and leukocytes, promoting metastasis. VLA-4 is expressed under physiological conditions in different subtypes of leukocytes but is also found in tumor cells. VLA-4 is able to bind to its ligand VCAM-1 expressed by activated endothelium, thereby mediating adhesion and a subsequent transmigration of tumor cells [49,50]. Regarding anticoagulation, LMWHs might be beneficial in removing the platelet aggregate and disrupting the formation of tumor thrombi, exposing cancer cells to immune cell recognition as nonself antigens and thus enhancing the immune system’s antitumor efficacy. LMWHs also inhibit CXCR4 in breast cancer mouse models [51], while CXCR4 inhibition also mitigates desmoplasia and enhances T cell infiltration, rendering metastatic breast cancer more susceptible to immunotherapy [52]. Interestingly, CXCR4 inhibition combined with anti-PD-1 checkpoint inhibitors has recently shown encouraging results in PDAC mouse models, and clinical trials are underway addressing this potential [53]. It could be presumed that a combination of standard chemotherapy, aiming to deplete regulatory T cells with LMWHs as CXCR4 inhibitors and checkpoint inhibitors, would be beneficial in terms of both tumor lethality and the abrogation of the immunosuppressive TME. 

### 3.3. Effects on Angiogenesis/Tumor Vasculature

Angiogenesis is a defining prerequisite for tumor growth beyond a specific size, as neovascularization is substantial for oxygen and nutrient support [54,55]. Mainly induced by hypoxic conditions in the TME, angiogenesis is a multistep process that involves growth factors and cytokines, with cancer and immune cells also playing a supportive role [56]. Hypoxia is a prominent feature that can induce angiogenesis by activating the hypoxia-inducible factor (HIF) pathway, whereas hypoxia-related and unrelated changes in oncogenes and tumor suppressor genes are implicated in proangiogenic signaling in cancer [57,58,59].

Pancreatic cancer angiogenesis is a decisive phenomenon. It is closely related to the hypercoagulable state observed even in early precancerous lesions, such as pancreatic intraepithelial neoplasia (PanIN) and intraductal papillary mucinous neoplasms (IPMNs), which are correlated with high levels of tissue factor (TF) expression [60], which in turn results in the activation of the coagulation cascade and the binding of TF to activated factor VII (FVIIa) and the formation of the TF–FVIIa complex [61,62]. TF initiates angiogenesis via both a) a clotting-dependent mechanism where thrombin formation and fibrin deposition support angiogenesis and TF induces VEGF expression, and b) a clotting-independent mechanism in which the TF–FVIIa complex activates proangiogenic protease-activated receptor 2 (PAR-2) [63,64]. Of note, VEGF enhances TF formation in endothelial cells in a positive feedback mechanism [65], whereas PAR-2 phosphorylates the TF cytoplasmic domain, thus abrogating the negative regulation usually exerted and resulting in the autocrine constitutive activation of PAR-2 and downstream angiogenic signaling [66,67].

VEGF, also activated by hypoxia-inducible factor-1a (HIF-1a) and released by tumor cells, dominates this extremely complex mechanism, while other proangiogenic features include angiopoietin-2, fibroblast growth factor (FGF), platelet-derived growth factor (PDGF), transforming growth factor-β (TGF-β), and interleukin-8 (IL-8) [54]. Tumor angiogenesis is sustained by a variety of overlapping interactions, ultimately resulting in the formation of abnormal vessels with increased permeability that in turn further promote desmoplasia and the attraction of immune-suppressive cells in the TME but also corroborate the critical steps during metastasis formation and the resistance of PDAC to therapy [56,68].

While primarily used in the treatment and prevention of venous thrombosis, LMWHs have been found to impair tumor neoangiogenesis, as shown in Figure 4. This effect is primarily based on their ability to prompt the prolonged release of tissue factor pathway inhibitor (TFPI) from binding sites on endothelial cells [69,70]. TFPI acts by inhibiting the TF–FVIIa complex and the subsequent activation of FX; it, therefore, disrupts the formation of thrombin and the activation of PAR-2, resulting in the abrogation of proangiogenic signaling [71,72]. TFPI interacts specifically with thrombospondin-1 (TSP-1) released by the alpha granules of platelets, and it resides in the ECM [73]. TSP-1 plays an antiangiogenic role by a) promoting the increased apoptosis of endothelial cells and b) by directly or indirectly inhibiting VEGF and other growth factors, e.g., FGF and PDGF [74,75,76,77]. 

### 3.4. Effects on Immune-Suppressive/Therapy-Resistant TME

Hypoxia has a significant role in the establishment of immune suppression; apart from inducing neoangiogenesis, hypoxia prompts the release of specific cytokines that promote the recruitment of regulatory T cells (Tregs), polarization of tumor-associated macrophages (TAMs) toward the immunosuppressive tumor-promoting M2 phenotype, expression of immune checkpoints, and downregulation of dendritic cells (DCs) maturation that results in reduced antigen presentation [56]. HIF-1a drives the differentiation of myeloid-derived suppressor cells (MDSCs) to immunosuppressive TAMs, whereas MDSCs in the TME are also responsible for tumor-associated antigen-specific T cell tolerance [78,79].

VEGF has a defining role in immunosuppression by indirectly enhancing hypoxic conditions as a result of an abnormal vasculature. VEGF also acts directly on immune cells within the TME and causes an increase of Tregs, MDSCs, and TAMs as well as the inhibition of DCs and antitumor TAMs [80].

Cancer-associated fibroblasts (CAFs) represent a dominant cell population in the PDAC microenvironment; once activated by cytokines produced by tumor cells, CAFs are responsible for excessive ECM protein production (mainly collagen, fibronectin, and matrix metalloproteinases MMPs) and desmoplasia as well as cytokine and growth factor release [81,82]. Growth factors (TGF-β, PDGF), in turn, assist in tumor progression but also provide an autocrine mechanism by which CAFs are constitutively activated [83,84]. A subpopulation of CAFs associated with the high expression of fibroblast activation protein (FAP) has immunosuppressive effects mediated through the activation of the CXCR4 receptor on effector T cells (Teff) [85], while desmoplasia is considered a significant barrier to adequate drug delivery within the TME [86].

As already described, LMWHs inhibit VEGF via TFPI release and alleviate hypoxia in the TME, thus downregulating the expression of HIF-1a and further driving the tumor toward normoxia. LMWHs also inhibit CXCR4, which potentially results in promoting the activity of Teff cells. It is possible that LMWHs also reduce TGF-β expression and the formation of fibrin monomers similar to the effect of enoxaparin in the breast cancer model [17], which could have a substantial effect on disrupting the perpetual autocrine CAF activation and ultimately abrogating desmoplasia, thus allowing both immune cells and chemotherapy molecules to penetrate the TME and promote antitumor efficacy. The above is summarized in Figure 5. 

## 4. Conclusions

Pancreatic adenocarcinoma remains one of the most challenging neoplasms, exhibiting great treatment hurdles and a dismal long-term overall survival. This is the result of specific features found in PDAC, including a highly desmoplastic stromal TME that acts by limiting effective drug delivery to malignant cells and by suppressing the host immune system to escape cancer cell elimination. Current research focuses on a multitarget approach that utilizes chemotherapy, immune checkpoint inhibitors, and a variety of adjuncts. These are considered to affect different sites within the TME in order to promote effective drug delivery and the normalization of the TME. This could reverse the elevated immunosuppression and allow immune cells to recognize and destroy the malignant ones. Our hypothesis is based on the ability of LMWHs to counteract many of the features described above that characterize the TME by promoting vascular normalization and the reprogramming of the microenvironment to obtain a more immune-supportive type. LMWHs are safely used in the prevention and treatment of VTE, which in PDAC is a significant factor associated with disease-related mortality. However, LMWHs are also attributed to various nonanticoagulant activities that can be exploited to target elements in the TME. Achieving this would allow both efficient chemotherapeutic drug delivery as well as effective immune cells that would infiltrate the tumor site, ultimately resulting in ameliorating the treatment efficacy and outcomes for patients.

## Figures and Tables

**Figure 1 cancers-12-00432-f001:**
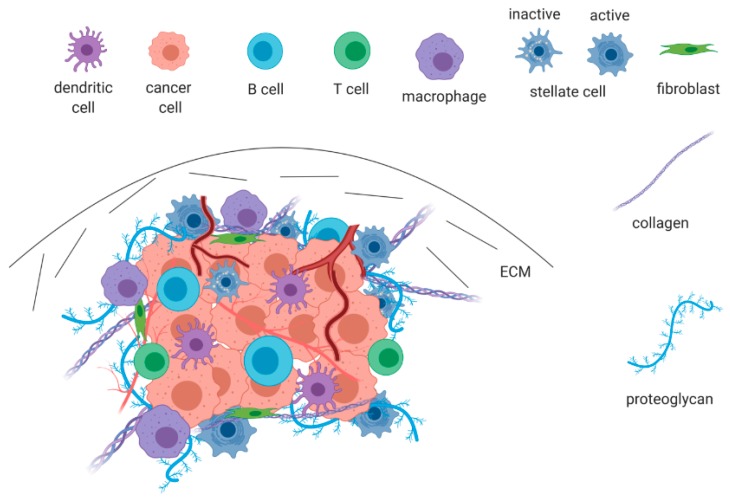
The pancreatic tumor microenvironment (TME). A complex ensemble of tumor cells, mesenchymal cells, inflammatory and immune cells, abnormal vascularity, and an excess of extracellular matrix (ECM). Hypoxia and excessive desmoplasia are the main features driving neoangiogenesis, immune suppression, and resistance to therapy.

**Figure 2 cancers-12-00432-f002:**
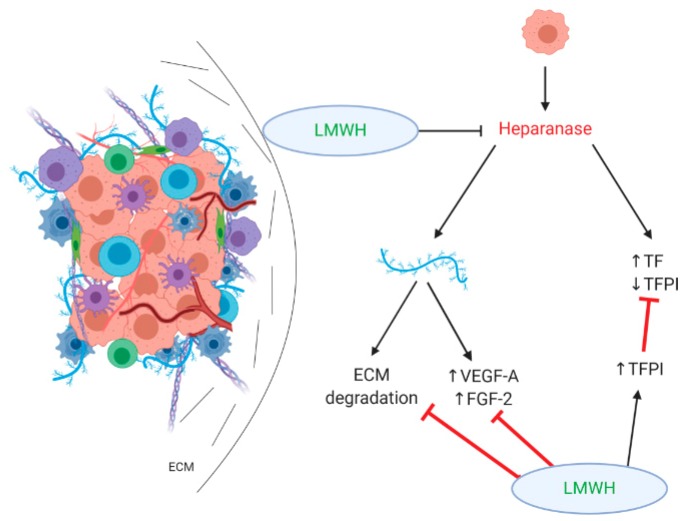
The effect of low molecular weight heparins (LMWHs) on heparan sulfate proteoglycans/heparanase. LMWHs antagonize the activity of heparanase through increased tissue factor pathway inhibitor (TFPI) release and the inhibition of vascular endothelial growth factor A (VEGF-A) and fibroblast growth factor 2 (FGF-2). They also inhibit the degradation of extracellular matrix (ECM), resulting in reducing desmoplasia in the TME.

**Figure 3 cancers-12-00432-f003:**
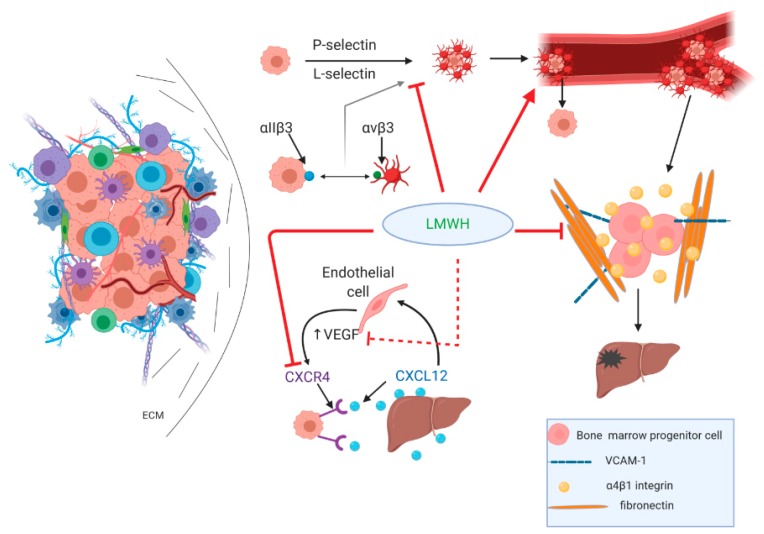
LMWHs inhibit metastasis formation. LMWHs inhibit P- and L-selectin and the integrin-mediated formation of tumor thrombi and disrupt the cellular interactions required for the metastatic homing of tumor cells by inhibiting VLA-4/vascular cell adhesion molecule (VCAM) interplay. The C-X-C motif chemokine ligand 12 (CXCL12)/C-X-C chemokine receptor type 4 (CXCR4) axis is blocked by both the direct inhibition of CXCR4 and the indirect downregulation of CXCR4 expression by tumor cells through vascular endothelial growth factor (VEGF) inhibition.

**Figure 4 cancers-12-00432-f004:**
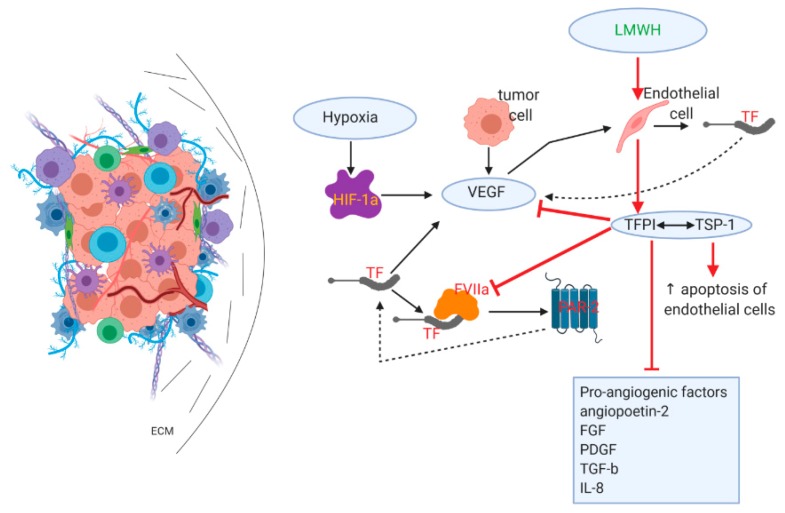
LMWHs reduce angiogenesis/tumor vasculature. LMWHs directly cause the release of TFPI from endothelial cells. TFPI inhibits the TF and activated factor VII (TF–FVIIa) complex, resulting in reduced protease-activated receptor 2 (PAR-2) activation and the elimination of angiogenic signaling. TFPI also inhibits a variety of proangiogenic factors, while interacting with thrombospondin-1 (TSP-1) causes an increase in endothelial cell apoptosis.

**Figure 5 cancers-12-00432-f005:**
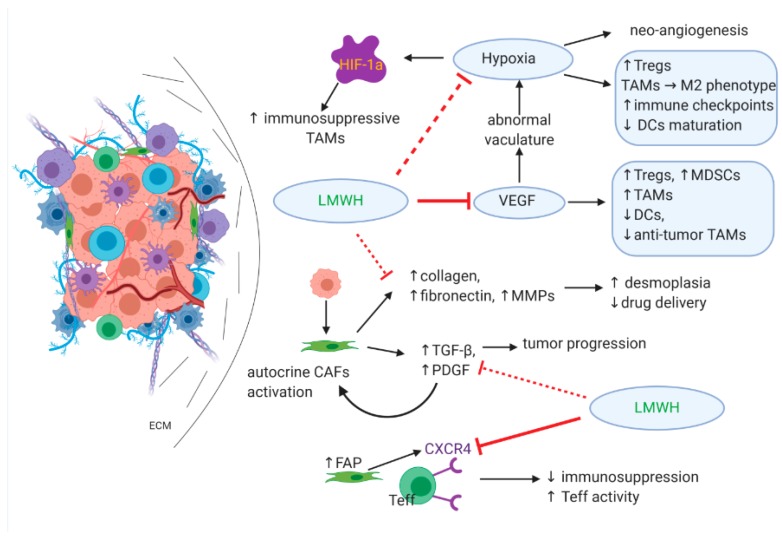
The effect of LMWHs on the immune-suppressive/therapy-resistant TME. The ability of LMWHs to reduce hypoxia directly or via VEGF inhibition results in normalizing tumor vascularity and reversing the immunosuppressive TME to an immune reactive one. This is further supported by CXCR4 inhibition, which promotes effector T cell activity. LMWHs also reduce fibroblast activation and, therefore, reduce desmoplasia and augment drug delivery in the TME. This results in both tumor cell death and the release of antigens, subsequently activating antitumor immunity.

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
