# Peer review of "Mechanisms of the Antitumor Activity of Low Molecular Weight Heparins in Pancreatic Adenocarcinomas"

_cancers, 2020, doi:10.3390/cancers12020432_

Round 1

Reviewer 1 Report

The paper describes a review of antitumor activity of low molecular weight heparins in pancreatic adenocarcinoma (PDAC). The title is attractive, although there are many problems as a review article.

The some contents described in the text is not correct.

For example,

Page 2, lines 22-23, In the escape phase, cancer cells are capable of modifying their characteristics, by expressing a lower antigen load, which helps them grow and eventually metastasize. (I think there is lack of necessary information.)

Page 3, lines 4-5, The presence of desmoplasia is a hallmark in PDAC pathogenesis and progression and can be recognized at both the primary and metastatic tumor sites. (Desmoplasia is found in the primary site but not usually in the metastatic site.)

The described contents might provide readers incorrect images.

For example,

Page 1, lines 34-36, Immune checkpoint inhibitors act by triggering the host immune system to fight against tumors by exploiting proteins expressed on T lymphocytes and tumor cells.

Page 2, lines 24-26, The basis of an immune response is the antigen presentation and the subsequent T cell activation, a process mainly mediated by cytokines interacting with cell receptors to induce sufficient anti-tumor activity.

There is often lack of contents necessary to describe as a useful review article.

For example,

Page 2, lines 28-29, Unfortunately, cancer cells can evolve to form primary and metastatic tumors despite the presence of an efficient immune system, mainly by exploiting a variety of mechanisms that enable them to avoid annihilation, while tumor growth can also be favored by differentiated immune cells.

Page 3, lines 19-21, Of the latter, immunosuppressive regulatory T cells (Tregs) appear in abundance compared to the fewer effector T cells and dendritic cells, while the presence of natural killer cells is unusual. (also no suitable citations)

Page 4, the last line – page 5, line 2, Also they serve as storage of growth factors, chemokines and enzymes in TME and regulate the availability and action of these molecules. (How does HS serve as a storage of these factors?)

Page 6, lines 12-13, LMWHs inhibit P- and L-selectin and VLA-4/VCAM mediated cellular interactions, thereby decreasing metastatic formation. (Why? How are the mechanisms?)

The citations are not suitable.

For example,

Page 2, lines 15-23, no suitable citations about immune editing

The tumor microenvironment is often different among tissue types and also sometimes different dependent on the species even in the same tissue. These differences are very important, especially in development of cancer therapy. Although, this review did not separate findings obtained from different species (i.e., human and mice). This style might be difficult to provide high-quality information to readers in this field. In addition, the described contents in the last part of several sections were not known findings but the authors’ opinions without any evidence in PDAC. The positive constructive opinions are important but we hope effective treatment evidence or failed treatment causes for PDAC in this review.

Author Response

We would like to thank the reviewer for the apt comments and constructive remarks. Below we provide our detailed answers to each comment one-by-one to ensure the clarity of our statements

The some contents described in the text is not correct.

For example,

Page 2, lines 22-23, In the escape phase, cancer cells are capable of modifying their characteristics, by expressing a lower antigen load, which helps them grow and eventually metastasize. (I think there is lack of necessary information.)

Thank you for the comment, we have modified the text and have rephrased it as: “In the escape phase, cancer cells are capable of modifying their characteristics with a loss of antigenicity and/or a loss of immunogenicity. Also, malignant cells can gain additional immunosuppressive properties, such as expression of PD-L1 or secretion of suppressive cytokines which results in additional reduction of their immunogenicity.” We have also added the citation: Gonzalez, H.; Hagerling, C.; Werb, Z. Roles of the immune system in cancer: From tumor initiation to metastatic progression. Genes Dev. 2018, 32, 1267–1284

Page 3, lines 4-5, The presence of desmoplasia is a hallmark in PDAC pathogenesis and progression and can be recognized at both the primary and metastatic tumor sites. (Desmoplasia is found in the primary site but not usually in the metastatic site.)

We thank the reviewer for the comment. We have altered the sentence as: “The presence of desmoplasia is a hallmark in PDAC pathogenesis and progression and can be recognized mainly at the primary tumor site”

The described contents might provide readers incorrect images.

For example,

Page 1, lines 34-36, Immune checkpoint inhibitors act by triggering the host immune system to fight against tumors by exploiting proteins expressed on T lymphocytes and tumor cells.

We thank the reviewer for the comment hence we have further analyzed the role of checkpoint inhibitors : “Tumor antigens are presented to T-cells by antigen-presenting-cells (APCs) and T-cell receptors, providing the primary signal for activating T-cells. Complete T-cell activation and expansion requires additional stimulatory signals whereas many co-receptors act as negative modulators of immune response at various checkpoints. The CTLA-4 is induced in T-cells at the time of their initial response to antigen, is transported to the cell surface where it binds to B7 and results in specific T-cell inactivation. On the other hand, the PD-1/PD1-L1 pathway is involved at later stages where activated T-cells up-regulate PD-1 and inflammatory signals stimulate the expression of PD1-L1s resulting in reduced activity of T-cells. This mechanism is utilized by tumor cells in order to escape the immune system response. Monoclonal antibodies that inhibit the action of either CTLA-4 or PD1/PD1-L1 enhance the activation of T lymphocytes, increase cytotoxic T-cell activity by expanding T-cell activation and proliferation”. We have also added the citation: Webb, E.S.; Liu, P.; Baleeiro, R.; Lemoine, N.R.; Yuan, M.; Wang, Y.-H. Immune checkpoint inhibitors in cancer therapy. J. Biomed. Res. 2018, 32, 317–326, doi:10.7555/JBR.31.20160168

Page 2, lines 24-26, The basis of an immune response is the antigen presentation and the subsequent T cell activation, a process mainly mediated by cytokines interacting with cell receptors to induce sufficient anti-tumor activity.

We thank the reviewer for the comment.We have altered the text to provide more accuracy and clarity as shown above: “The principle mode of action of the immune system is recognition by cells of the innate immune system, the release of various cytokines, complement activation and concomitant phagocytosis. Tumor development can be controlled by cytotoxic innate and adaptive immune cells but as the tumor evolves from neoplastic tissue to detectable tumors, cancer cells utilize different mechanisms that recapitulate peripheral immune tolerance in order to survive, grow and metastasize.” We have also added the citation: Pio, R.; Ajona, D.; Ortiz-Espinosa, S.; Mantovani, A.; Lambris, J.D. Complementing the cancer-immunity cycle. Front. Immunol. 2019, 10.

There is often lack of contents necessary to describe as a useful review article.

For example,

Page 2, lines 28-29, Unfortunately, cancer cells can evolve to form primary and metastatic tumors despite the presence of an efficient immune system, mainly by exploiting a variety of mechanisms that enable them to avoid annihilation, while tumor growth can also be favored by differentiated immune cells.

Thank you for the remark, we also believe that more information should be provided, therefore the text has been changed to: “Unfortunately, cancer cells can evolve to form primary and metastatic tumors despite the presence of an efficient immune system. Tumor cells evade the immune system attack by using two main strategies: avoiding the immune recognition and instigating an immunosuppressive TME. Firstly,cancer cells rapidly decrease the expression of tumor antigens on the cell surface, thus avoiding the recognition by cytotoxic T cells. Furthermore, in the second, cancer cell-derived factors instigate an immune-tolerant TME by (a) secretion of suppressive molecules (b) expression of inhibitory checkpoint molecules and (c) induction of the recruitment of TAMs, MDSCs, and Tregs”. We have also added the bibliography: H.; Hagerling, C.; Werb, Z. Roles of the immune system in cancer: From tumor initiation to metastatic progression. Genes Dev. 2018, 32, 1267–1284

Page 3, lines 19-21, Of the latter, immunosuppressive regulatory T cells (Tregs) appear in abundance compared to the fewer effector T cells and dendritic cells, while the presence of natural killer cells is unusual. (also no suitable citations)

We thank the reviewer for the remark hence we have made the modifications required to ensure the required articulacy: “Tregs have the capacity to infiltrate highly in various tumor types such as pancreatic cancerand advanced cancers often display higher levels of Treg accumulation.   Tregs elevation inside the tumors is performed via TGFb that induces the differentiation of naïve CD4+ T cells into Tregs. In addition, chemokines produced by the tumor play an important role such as CCL22 that is able to stimulate CCR4. Tregs suppress T cell immune responses of antigen-presenting cells, including DCs and macrophages. , Tregs can kill effector T cells within tumors through the FasL-Fas signaling pathway as well as via granzyme B- and perforin-mediated cytotoxicity, through reduction of IL-2 and differentiation/maturation of DC”. We have also added the citation: Wang, H.; Franco, F.; Ho, P.C. Metabolic Regulation of Tregs in Cancer: Opportunities for Immunotherapy. Trends in Cancer 2017, 3, 583–592.

Page 4, the last line – page 5, line 2, Also they serve as storage of growth factors, chemokines and enzymes in TME and regulate the availability and action of these molecules. (How does HS serve as a storage of these factors?)

Thank you for the remark, we have changed the sentences describing that HS serve as a storage of these factors. The alterations are the above: “ The heparan sulfate chains due to their broad structural diversity are able to bind and interact with a wide variety of proteins such as growth factors , chemokines and enzymes in TME regulating the availability and action of these molecules”. We have also added the bibliography Sarrazin, S.; Lamanna, W.C.; Esko, J.D. Heparan sulfate proteoglycans. Cold Spring Harb. Perspect. Biol. 2011, 3, 1–33, doi:10.1101/cshperspect.a004952.

Page 6, lines 12-13, LMWHs inhibit P- and L-selectin and VLA-4/VCAM mediated cellular interactions, thereby decreasing metastatic formation. (Why? How are the mechanisms?)

We thank the reviewer for the comment and we have embodied to the text a brief description of the mechanisms:” Selectins are vascular cell adhesion molecules that recognize specific sialyl Lewisx/a carrying glycoproteins that exist on normal leukocytes and endothelium. Tumor cell surface molecules carrying sialyl Lewisx/a can also be recognized from selectins mediating tumor cell interactions with platelets, leukocytes, promoting metastasis. VLA-4 is expressed under physiological conditions on different subtypes of leukocytes, but is also found on tumor cells. VLA-4 is able to bind to its ligand VCAM-1 expressed by activated endothelium thereby mediating adhesion and a subsequent transmigration of tumour cells”. We have also added the citations: Borsig, L.; Wong, R.; Hynes, R.O.; Varki, N.M.; Varki, A. Synergistic effects of L- and P-selectin in facilitating tumor metastasis can involve non-mucin ligands and implicate leukocytes as enhancers of metastasis. Proc. Natl. Acad. Sci. U. S. A. 2002, 99, 2193–8, doi:10.1073/pnas.261704098.

Schlesinger, M.; Roblek, M.; Ortmann, K.; Naggi, A.; Torri, G.; Borsig, L.; Bendas, G. The role of VLA-4 binding for experimental melanoma metastasis and its inhibition by heparin. Thromb. Res. 2014, 133, 855–862, doi:10.1016/j.thromres.2014.02.020.

The citations are not suitable.

For example,

Page 2, lines 15-23, no suitable citations about immune editing

Thank you for this specific comment. We understand the requirement for more suitable citations therefore we have further modified the references. For example we have included the reference below Beatty, G.L.; Gladney, W.L. Immune escape mechanisms as a guide for cancer immunotherapy. Clin. Cancer Res. 2015, 21, 687–692.

The tumor microenvironment is often different among tissue types and also sometimes different dependent on the species even in the same tissue. These differences are very important, especially in development of cancer therapy. Although, this review did not separate findings obtained from different species (i.e., human and mice). This style might be difficult to provide high-quality information to readers in this field. In addition, the described contents in the last part of several sections were not known findings but the authors’ opinions without any evidence in PDAC. The positive constructive opinions are important but we hope effective treatment evidence or failed treatment causes for PDAC in this review

We thank the reviewer for his remarks and his recommendations for the improvement of the manuscript. As for the comment “The tumor microenvironment is often different among tissue types and also sometimes different dependent on the species even in the same tissue. These differences are very important, especially in development of cancer therapy”, we have added at section 3 (The pancreatic adenocarcinoma microenvironment) the sentence “Τhe fact that tumor microenvironment is a dynamic entity may account for the diversity observed in different tissues; however, main features such as desmoplasia, abnormal vascularity and fibroblast activation are almost always present” with the appropriate citations. Regarding the rest of the comments the possible mechanisms referred in this manuscript have been described at experiments performed mainly to human cell lines and mice in pancreas and other tissues. We provide the citations where these experiments are thoroughly described and we used as guidelines for the composition of this text. We tried to provide a theoretical basis of our approach and point out mechanisms by which the addition of LMWH could interfere with tumor development and contribute to the normalization of the microenvironment, ultimately leading to higher activity of the abovementioned drug combination.

Reviewer 2 Report

This review on “Mechanisms of low molecular weight heparins anti-tumor activity in pancreatic adenocarcinomas” summarizes available evidence and hypothesis regarding the potential antitumor activity of LMWH in pancreatic adenocarcinoma.

The manuscript is well written and interesting and describes the various mechanisms by which LMWH might interfere with the tumor development.

However, I have some minor concerns regarding the manuscript that should be addressed by the authors :

-In the title, replace “molecucar” by “molecular”

- at the end of the introduction section, the autors state that “over 20% of patients suffer from venous thromboembolism (VTE) during disease” : please replace “during disease” by “during the course of their disease”; it would be interesting to add a reference regarding the incidence of VTE in pancreatic cancer patients and the authors might cite the results of the recent prospective, observational multicenter BACAP-VTE study (Frere C, Bournet B, Gourgou S, Fraisse J, Canivet C, Connors JM, Buscail L, Farge D; BACAP Consortium. Incidence of Venous Thromboembolism in Patients with Newly Diagnosed Pancreatic Cancer and Factors Associated With Outcomes. Gastroenterology. 2019 Dec 13. pii: S0016-5085(19)41921-5. doi: 10.1053/j.gastro.2019.12.009. [Epub ahead of print]) which demonstrate that VTE occurs in 20.79% of patients with a new diagnosis of PDAC of any stage and is associated with a shorter progression free survival and a shorter overall survival.

-at the end of the introduction section, “LMWHs have replaced UFH in the management of VTE and have been established as the recommended anticoagulant for cancer patients”; I would change the sentence for “LMWHs are recommended as first line option for the treatment and primary prophylaxis of VTE  in pancreatic cancer patients”. Indeed, the recent ITAC guidelines recommend the use of LMWH for primary pharmacological prophylaxis of VTE in ambulatory patients with locally advanced or metastatic pancreatic cancer receiving systemic anticancer therapy and who have a low risk of bleeding with a grade 1B mainly based on the results of the PROSPECT-CONKO 004 and FRAGEM trial.

Please add the three following references:

Farge D, Frere C, Connors JM, Ay C, Khorana AA, Munoz A, Brenner B, Kakkar A, Rafii H, Solymoss S, Brilhante D, Monreal M, Bounameaux H, Pabinger I, Douketis J; International Initiative on Thrombosis and Cancer (ITAC) advisory panel. 2019 international clinical practice guidelines for the treatment and prophylaxis of venous thromboembolism in patients with cancer. Lancet Oncol. 2019 Oct;20(10):e566-e581.

Maraveyas, A.; Waters, J.; Roy, R.; Fyfe, D.; Propper, D.; Lofts, F.; Sgouros, J.; Gardiner, E.; Wedgwood, K.; Ettelaie, C.; et al. Gemcitabine versus gemcitabine plus dalteparin thromboprophylaxis in pancreatic cancer. Eur. J. Cancer Oxf. Engl. 1990 2012, 48, 1283–1292.

Pelzer, U.; Opitz, B.; Deutschinoff, G.; Stauch, M.; Reitzig, P.C.; Hahnfeld, S.; Müller, L.; Grunewald, M.; Stieler, J.M.; Sinn, M.; et al. Efficacy of Prophylactic Low-Molecular Weight Heparin for Ambulatory Patients With Advanced Pancreatic Cancer: Outcomes From the CONKO-004 Trial. J. Clin. Oncol. Off. J. Am. Soc. Clin. Oncol. 2015, 33, 2028–2034.

-at the end of the introduction section, “There have been accumulating data suggesting an increase in overall survival (OS) of cancer patients receiving LMWHs.” I would modulate the sentence. Indeed, the FRAGEM and PROSPECT-CONKO 004 failed to demonstrate a benefit of LMWH on overall survival in PC patients.

-page 4 : “The role of LMWHs in pancreatic cancer”. I would change the sentence at the beginning of the paragraph : “Beyond their ability in decreasing VTE occurrence in pancreatic cancer patients, LMWHs may contribute to increase survival by affecting tumor progression, metastasis formation and angiogenesis…  

Author Response

Initially, we would like to thank the reviewer for his very kind and fruitful comments that will be beneficial for the current manuscript.

In the title, replace “molecucar” by “molecular”

We thank the reviewer for the remark and we have made the change to the word

at the end of the introduction section, the autors state that “over 20% of patients suffer from venous thromboembolism (VTE) during disease” : please replace “during disease” by “during the course of their disease”; it would be interesting to add a reference regarding the incidence of VTE in pancreatic cancer patients and the authors might cite the results of the recent prospective, observational multicenter BACAP-VTE study (Frere C, Bournet B, Gourgou S, Fraisse J, Canivet C, Connors JM, Buscail L, Farge D; BACAP Consortium. Incidence of Venous Thromboembolism in Patients with Newly Diagnosed Pancreatic Cancer and Factors Associated With Outcomes. Gastroenterology. 2019 Dec 13. pii: S0016-5085(19)41921-5. doi: 10.1053/j.gastro.2019.12.009. [Epub ahead of print]) which demonstrate that VTE occurs in 20.79% of patients with a new diagnosis of PDAC of any stage and is associated with a shorter progression free survival and a shorter overall survival.

Thank you for the apt remarks and the recommendation for a specific reference you have shown. It is very interesting and important for our review and we have already added it to our bibliography

-at the end of the introduction section, “LMWHs have replaced UFH in the management of VTE and have been established as the recommended anticoagulant for cancer patients”; I would change the sentence for “LMWHs are recommended as first line option for the treatment and primary prophylaxis of VTE in pancreatic cancer patients”. Indeed, the recent ITAC guidelines recommend the use of LMWH for primary pharmacological prophylaxis of VTE in ambulatory patients with locally advanced or metastatic pancreatic cancer receiving systemic anticancer therapy and who have a low risk of bleeding with a grade 1B mainly based on the results of the PROSPECT-CONKO 004 and FRAGEM trial.

Please add the three following references:

Farge D, Frere C, Connors JM, Ay C, Khorana AA, Munoz A, Brenner B, Kakkar A, Rafii H, Solymoss S, Brilhante D, Monreal M, Bounameaux H, Pabinger I, Douketis J; International Initiative on Thrombosis and Cancer (ITAC) advisory panel. 2019 international clinical practice guidelines for the treatment and prophylaxis of venous thromboembolism in patients with cancer. Lancet Oncol. 2019 Oct;20(10):e566-e581.

Maraveyas, A.; Waters, J.; Roy, R.; Fyfe, D.; Propper, D.; Lofts, F.; Sgouros, J.; Gardiner, E.; Wedgwood, K.; Ettelaie, C.; et al. Gemcitabine versus gemcitabine plus dalteparin thromboprophylaxis in pancreatic cancer. Eur. J. Cancer Oxf. Engl. 1990 2012, 48, 1283–1292.

Pelzer, U.; Opitz, B.; Deutschinoff, G.; Stauch, M.; Reitzig, P.C.; Hahnfeld, S.; Müller, L.; Grunewald, M.; Stieler, J.M.; Sinn, M.; et al. Efficacy of Prophylactic Low-Molecular Weight Heparin for Ambulatory Patients With Advanced Pancreatic Cancer: Outcomes From the CONKO-004 Trial. J. Clin. Oncol. Off. J. Am. Soc. Clin. Oncol. 2015, 33, 2028–2034.

We thank the reviewer for the comment and the interesting papers. We have made the change and added the references

at the end of the introduction section, “There have been accumulating data suggesting an increase in overall survival (OS) of cancer patients receiving LMWHs.” I would modulate the sentence. Indeed, the FRAGEM and PROSPECT-CONKO 004 failed to demonstrate a benefit of LMWH on overall survival in PC patients.

We thank the reviewer for the remark, indeed there was a misconception to our meaning. The citation we have provided is in regard to the benefit to VTE patients and not cancer patients suffering from VTE, therefore we have withdrew the sentence.

page 4 : “The role of LMWHs in pancreatic cancer”. I would change the sentence at the beginning of the paragraph : “Beyond their ability in decreasing VTE occurrence in pancreatic cancer patients, LMWHs may contribute to increase survival by affecting tumor progression, metastasis formation and angiogenesis…  

We thank the reviewer for the remark which we have included to our manuscript.

Round 2

Reviewer 1 Report

This reviewer said there were several problems in the original paper and some of the problem points were indicated as “for example” that means all problems points were not raised. In the revised paper, the authors improved the points indicated. However, any other points (not indicated in the original review) have not been improved yet.

This reviewer believes that Review article should tell the current truth to readers as possible. There are still problems in this report. As an useful review article, the enough and correct information is not provided in this article, especially in immunological part, general PDAC pathology, and PDAC TME. Since this reviewer is not a mentor or a coauthor for the author of this paper, it is not necessary to point out all the individual problem points. 

Author Response

We would like to thank the reviewer for his comments. We will agree with the Academic Editor that we do not need to analyze in depth the immunological part, general PDAC pathology and PDAC TME as this review does not aim to extensively elucidate these mechanisms but provide information on how LMWHs can contribute to more efficient therapies of PDAC.

Reviewer 2 Report

The authors have now addressed all concerns. I have no additional comment.

Author Response

We would like to thank the reviewer for his very fruitful and kind comments and we are very satisfied that our answers covered his concerns.